# Oncological and Functional Outcomes for Horizontal Glottectomy: A Systematic Review

**DOI:** 10.3390/jcm12062261

**Published:** 2023-03-14

**Authors:** Matteo Fermi, Alfredo Lo Manto, Cecilia Lotto, Giulia Cianci, Francesco Mattioli, Daniele Marchioni, Livio Presutti, Ignacio Javier Fernandez

**Affiliations:** 1Department of Otorhinolaryngology Head and Neck Surgery, IRCCS Azienda Ospedaliero-Universitaria di Bologna, Policlinico S. Orsola-Malpighi, 40138 Bologna, Italy; 2Department of Medical and Surgical Sciences (DIMEC), Alma Mater Studiorum, Università di Bologna, 40126 Bologna, Italy; 3Department of Otorhinolaryngology Head and Neck Surgery, University Hospital of Modena, 41125 Modena, Italy

**Keywords:** laryngeal cancer, head and neck cancer, horizontal glottectomy, oncological and functional outcomes

## Abstract

Horizontal glottectomy (HG) is a particular type of partial laryngectomy indicated for exclusive glottic tumor with anterior commissure involvement. The purpose of this study is to systematically review the literature about functional and oncological outcome of HG. This systematic review adhered to the recommendations of the PRISMA (Preferred Reporting Items of Systematic Reviews and Meta-analysis) 2009 guidelines. Articles mentioning patients undergoing HG for laryngeal squamous cell carcinoma were included. A total of 14 articles were selected and reviewed from 19 identified. The whole study population consisted of 420 patients who underwent HG. Three hundred and thirty-nine patients out of 359 were staged as T1. The range of post-operative follow-up was 5 months to 10 years. Fifty-five recurrences were experienced, being local, regional and distant in 35, 12 and 8 patients, respectively. Laryngeal preservation rate was 93.6%. Nasogastrict tube was removed on average after 10.1 days. The tracheostomy was maintained for 11.3 days. Mean hospitalization lasted for 11.7 days. According to the results of this systematic review, HG is an oncologically safe surgical option for T1a–T1b glottic tumors with oncological outcomes comparable to other treatment. HG could be a good therapeutical choice whenever poor laryngeal exposure and/or patient’s refusal of radiotherapy are encountered, or when patient’s medical history represents a contraindication for radiation therapy.

## 1. Introduction

Oncological principles in the treatment of laryngeal cancer have evolved in recent years.

The goals in the treatment of patients with early-stage glottic laryngeal cancer (T1 and T2) are cure from the disease and preservation of laryngeal function to maximize quality of life.

Patients with such tumors should be considered for a single modality of therapy whenever possible without decreasing the chance for cure. Such single-modality options include transoral laser microsurgery (TLM), open partial horizontal laryngectomy (OHPL), or radiation therapy (RT) alone.

More specifically, the gold standards of care for T1a and T1b glottic tumors are both RT and TLM, but in the past few decades, a particular type of partial laryngectomy, the horizontal glottectomy (HG), has been described for treating tumors with exclusive glottic extension with anterior commissure (AC) involvement.

HG was first described by Calearo and Teatini in 1978 and belongs to the larger group of conservative laryngectomies. More specifically, it is a segmental resection of the glottis performed by two horizontal incisions: the lower through the cricothyroid membrane and the upper across the wings of the thyroid cartilage. The resulting defect is closed by approximating the cricoid to the thyroid remnants performing a cricothyropexy [1]. This technique involves the removal of the entire glottis, anterior commissure (AC) and both vocal folds, together with the lower half of thyroid cartilage. The technique also allows extension of the procedure to one arytenoid and/or ventricular band. This procedure is indicated in selected cases of bilateral vocal cord cancer with the extension on the AC. Vocal cord mobility should be normal, and there should be no subglottic or supraglottic extension [2]. With the advent of TLM and open partial laryngectomies (OPHL type II), this technique has gradually lost its importance. Transoral laser microsurgery is an emerging technique consisting of a resection conducted from the inside working outward. For this reason, those who use this technique assert that there is less damage to the structures that lie external to the tumor. Moreover, it is a technique characterized by lower morbidity and improved organ preservation. The postoperative complications are less common, and consequently the hospitalization is shorter. In particular, it is a useful first-line technique because it permits other treatments in patients with recurrent diseases. Other types of open partial laryngectomies are techniques characterized by more frequent postoperative complications that require longer hospitalization with a difficult rehabilitation management for the swallowing and phonation. In particular, the selection of patients based on age and comorbidities is of primary importance. On the other hand, this technique permits an excellent oncological outcome.

Probably for this reason, there are few published studies concerning the effectiveness of HG in terms of local tumor control and functional results. The purpose of this study is to systematically review the literature regarding functional and oncological outcomes of HG, in order to define the current state-of-the-art regarding this procedure and to allow for its selective use in the future.

## 2. Materials and Methods

### 2.1. Eligibility Criteria

This systematic review adhered to the recommendations of the PRISMA (Preferred Reporting Items of Systematic Reviews and Meta-Analyses) 2009 guidelines, to guarantee a scientific strategy of research to limit bias through a systematic assembly, critical appraisal, and synthesis of all of the relevant studies published on the chosen topic [3]. With the research question being focused on the indications, oncological and functional outcomes of HG for glottic laryngeal cancer, data from studies on patients that underwent HG were pooled for the review process.

### 2.2. Information Sources and Search

In March 2022, a computerized MEDLINE search was performed using the PubMed service of the U.S. National Library of Medicine (www.pubmed.org accessed on 3 March 2022) and Scopus database (www.scopus.com accessed on 3 March 2022), running the following search string: “horizontal glottectomy”. The initial search returned a total of 19 results. After electronically removing duplicates, titles and abstracts obtained were screened independently by three of the authors (A.L.M., C.L. and G.C.), who subsequently met and discussed disagreement on citation inclusion. The inclusion criteria for citations were set a priori to encompass as many articles as possible without compromising the validity of the results. Articles mentioning patients undergoing HG for laryngeal squamous cell carcinoma (LSCC) were included.

Exclusion criteria were articles written in languages other than English, French or Italian, or with a research subject clearly unrelated to HG for laryngeal primary tumor.

Afterwards, the selected full-text articles underwent a second screening by the same three authors. Full-text articles were considered regardless of their study design, in order not to miss any relevant data, and were included if: (1) they reported case-series of LSCC treated with HG according to Calearo and Teatini description; (2) they reported HG oncologic and/or functional outcomes; (3) they reported perioperative or postoperative complications; and/or (4) they reported a follow-up period [1]. Full texts reporting data from the same study population were excluded.

After including 3 articles by manual search on the references from the pooled full texts, the final number of articles included in the review was identified (Figure 1), and the main information was extracted and summarized in a database.

Statistical analysis was performed using SPSS software 16.0 version (SPSS Inc./IBM).

The results are expressed as mean (standard deviation—SD) for continuous variables with normal distribution, median (IQR) for continuous variables with non-normal distribution and as percentages for categorical variables.

### 2.3. Assessment of Quality and Risk of Bias

Two authors (CL and GC) evaluated the methodological quality of studies identified using the Oxford Centre for Evidence-Based Medicine Levels of Evidence. The risk of bias was assessed by assigning a score using the Methodological Index for Non-Randomized Studies (MINORS), an extensively validated instrument for literature assessment. For non-comparative studies, eight domains are assessed, scoring the items as 0 (not reported), 1 (reported but inadequate) or 2 (reported and adequate). The optimal score for non-comparative studies is therefore 16. For the purposes of this review, a value of 10 or below was considered to represent a high risk of bias.

## 3. Results

A total of 14 studies were included in this systematic review, as shown in the flow chart (Figure 1) [4,5,6,7,8,9,10,11,12,13,14,15,16,17,18].

Four (80%) were retrospective observational studies [4,5,6,12,14], while one [6] (20%) was cross-sectional. The remaining nine studies did not specify their study design.

The whole study population consisted of 420 patients (327 males, 31 females, 62 not specified) undergoing HG over 37 years (1973–2010). Histological type was laryngeal squamous cell carcinoma tumors, which was specified in 8 out of 14 studies. The mean reported age was 61.27 ± 13.65 years. The tumor’s histologic diagnosis was invasive carcinoma in 334 (79.5%) patients, carcinoma in situ, dysplasia and papillomatous lesion in 14 (3.3%), 9 (2.1%) and 2 (0.5%) patients respectively, while the histotype was not specified in the remaining 61 (14.5%) patients. The preoperative tumor staging was available in 11 out of the 14 studies as shown in Table 1. Most of those patients (339 patients out of 359) (94.4%) were staged T1 (T1a 38—11.2%; T1b 301—88.79%). Data regarding neck dissection were lacking in five studies [6,11,13,15,16], while considering the remaining studies, only 17 out of 322 patients underwent simultaneous neck dissection (5.3%). Data regarding status of margins were available only in eight studies [5,6,9,10,13,14,15,17], with positive, close or negative results in 13 (4.5%), 29 (10%) and 247 (85.5%) patients, respectively, out of 289 overall.

Nasogastric tube was removed on average after 10.1 days (range 3–42), while the tracheostomy was maintained for 11.3 (range 2–31) days. Overall, mean hospitalization lasted for 11.7 (range 3–37) days.

Eighteen (4.3%) patients underwent post-operative radiation therapy, based on the post-operative histopathologic features of the tumor.

The range of post-operative follow-up was available for all of the articles herein analyzed, being from 5 months to 10 years (45.4 ± 11.4). Overall, 55 (13.8%) recurrences were experienced, being local, regional and distant in 35 (8.8%), 12 (3%) and 8 (2%) patients, respectively. Among the patients who recurred, 27 underwent total laryngectomy, achieving an overall laryngeal preservation rate of 93.6%. The overall survival (OS) and disease-specific survival (DSS) data were available in eight [6,8,9,11,13,15,16,17] and nine studies [7,8,9,11,12,13,15,16,17], respectively, combining for a mean OS and DSS of 89% and 93.3%, respectively. The oncological outcome is summarized in Figure 2.

The reported functional vocal results were few and heterogenous. Four [5,7,8,18] out of 14 studies considering 169 patients, evaluated GRBAS scale, for perceptual voice estimation, resulting in an average G (expressed in all of the studies considered) of 2.38 ± 0.58. Maximum Phonation Time (MPT) and Voice Handicap Index-30 (VHI)19 were reported in five out of 14 studies [4,5,7,8,18]. The MPT average resulted 8.1 ± 1.7 s. The mean VHI average was 33.3 ± 4.3 [19]. Only three studies considered the perturbation analysis [7,8,18]. One study considered only F0 (median results 132.71) [4].

Regarding the swallowing analysis, only three studies performed the FEES [6,7,8], and in only one of them were the grade of premature spillage (4.77 ± 0.21), retention pooling (4.64 ± 0.27) and aspiration (4.85 ± 0.18) described. Other evaluation scales, such as MDADI8, Yanaghiara score [20] or Pearson and Leipzig scale [21,22] were reported in only one study (Analysis of Vocal and Swallowing Functions after Horizontal Glottectomy by Topaloğlu et al. for MDADI8 and Yanaghiara score, and Value of Open Horizontal Glottectomy in the Treatment for T1b Glottic Cancer with Anterior Commissure Involvement: Open Horizontal Glottectomy in T1B Glottic Cancer by Szyfter et al. for Pearson and Leipzig scale).

According to the MINORS score, most of the studies (57%) achieved a score above 10, while the remaining studies (43%) did not. The MINORS scores, indicating the risk of bias, are listed in Table 2. Only the six studies published before 2000 did not achieve a sufficient score. In fact, in most of them, the main outcomes were not adequately explained, and the criteria used to assess them were often unclear. For instance, oncologic and functional outcomes were sometimes reported without quantitative and validated measures (e.g., validated questionnaires, objective measures, etc.). Moreover, the follow-up of some of those study populations was not sufficiently long to allow the assessment of the main endpoint and possible treatment’s adverse events.

Conversely, in the more recent studies, the MINORS criteria were fulfilled. More specifically, the study aim was clear, as were the inclusion and exclusion criteria (e.g., preoperative staging, disease features, etc.) of the study cohort. The results of those studies were based on a median follow-up long enough to have reliable oncologic and functional outcomes (almost all those studies report a 5-year follow-up cut-off), and the methods for the outcomes’ evaluation were clearly explained.

Lastly, no prospective studies were available on the selected topic, and all of the studies herein analyzed are retrospective case-series.

## 4. Discussion

HG is a conservative surgical technique that allows the removal of bilateral glottic carcinomas with anterior commissure involvement while preserving swallowing status. Nowadays, the treatment modalities for these kinds of tumors are increasingly based on other types of organ preservation surgical technique, such as TLM or RT, while OPHL are considered in selected cases [23].

This is the first systematic review pooling the oncologic and functional results of the HG as a treatment modality for selected laryngeal carcinomas.

Few studies in the literature have focused on HG showing that this technique has never gained a widespread popularity.

Fourteen studies were included in this systematic review, collecting a study population of 401 patients. Most of these articles (12/14) reported oncological outcomes, although different follow-up periods and variables were considered. Clinical staging was reported in 11 studies: 301 of 359 patients were staged as T1b glottic tumors. OS and DSS were 89% and 93.3%, respectively, while the LPR was 93.6%.

One of the major problems in T1–T2 glottic tumors treated with TLM or RT is anterior commissure involvement.

AC’s anatomical peculiarity makes this laryngeal subsite a challenging area for adequate pretreatment evaluation (endoscopic and imaging workup with a high incidence of false negatives), a difficult area for surgical resection due to laryngeal exposure, and prone to technical radiotherapy issues at the air–tumor interface within the AC due to its variable degree of vascularization, ossification, and three-dimensional conformation further complicating its proper management [24].

When performing TLM, good exposure and visualization of the glottic plane, particularly of the anterior one third of the vocal folds and anterior commissure, are mandatory. In this context, an important role is played by the Laryngoscore. This score encompasses 11 parameters that predict the degree of laryngeal exposure before TLM. In the study of Piazza et al., cases with suboptimal exposure had more than twice the number of involved margins than ones with good laryngeal exposure [25,26]. Thus, whenever a suboptimal laryngeal exposure is encountered, an alternative therapeutic approach, such as RT or open laryngeal surgery (e.g., HG or OPHL) should be considered.

Accordingly, different data indicate results of disease-free survival and recurrence rates are poorer in patients with anterior commissure involvement who underwent a transoral laser cordectomy [24,27,28,29,30]. Lei et al. performed a retrospective study with a cohort of 201 patients with early glottic tumors treated with TLM showing a local recurrence (LR) and DSS of 30.8% and 82.5%, respectively, for early glottic tumors with AC involvement and of 16% and 96.5% for glottic tumors without anterior commissure involvement [31]. A recent meta-analysis by Zhou et al. revealed that early glottic cancer with AC involvement is more likely to have 2.39 times higher local recurrence compared to ACi-glottic tumors. In particular, the LR rate was 24.5% and 10.5% for AC involvement and non-involvement, respectively. Moreover, the laryngeal preservation rate for glottic tumors with AC involvement was 93.8%, compared to 97.6% for those without the involvement of the AC [32].

Therefore, adequate visualization of the entire glottis, particularly at the level of the AC, should always be regarded as an essential prerequisite for an effective TLM procedure.

The literature regarding radiotherapy is ambiguous with respect to the outcomes of laryngeal cancers with AC involvement. However, several studies reported worst local control in T1b with AC involvement tumors [33,34,35,36,37,38,39,40]. Mucha-Malecka et al. performed a retrospective study based on a group of 569 patients T1N0M0 laryngeal cancer treated with radiotherapy. In the univariate analysis, lesions with AC involvement were related to a significant decrease in LC rates and DSS rates (77% and 75%, respectively) [33]. Numerous authors have also observed that the involvement of the AC early glottic tumors was correlated with a two- to three-fold increase in local relapse risk [39,40].

OPHL type II, previously called supracricoid laryngectomy, is an open surgical therapy for early- or intermediate-stage laryngeal carcinoma. It differs from HG because the pexy is performed between the cricoid ring and the hyoid bone, and thus the supraglottic larynx is also sacrificed. Several studies show good oncological results for t1b glottic tumors invading the AC, with an optimal local control at 5-year follow-up [41,42,43]. However, the HG procedure maintains the supraglottis, reconstructs a symmetric glottis, and results in a higher degree of glottic closure. Furthermore, a smaller portion of the larynx is sacrificed, and the variation in organ length is less in HG than in OPHL II compared to the normal larynx. These aspects, although not proven, could allow for better phonation and swallowing than OPHL type II.

The functional outcomes considered in the studies included in this systematic review are very heterogeneous, probably because the functional variables considered over the years have varied with the new technologies available.

Advantages of TLM over alternative modalities include shorter hospitalization, lower morbidity, and superior functional outcomes compared to those of OPHL and shorter treatment time and less damage to healthy tissue compared to those of RT.

In the pooled study population of patients undergoing HG, the nasogastric tube was removed on average 10.1 days (range 3–42 days) after surgery, while the tracheostomy was maintained for 11.3 days. Overall, mean hospitalization lasted for 11.7 days (range: 3–37 days). Although HG has challenging postoperative management in comparison with TLM, these results appear to be similar to or better than those reported for OPHL II. Pinar et al. retrospectively evaluated 56 patients who underwent OPHL II A or OPHL II B [44]; the mean decannulation (MDT) time and nasogastric feeding tube (NGT) removal time were 11.43 (±2.03) and 16.79 (±3.51) days, respectively. Mesonella et al. also reported functional outcomes of 35 patients who underwent OPHL II. MDT and NGT removal were 21.8 (range: 14 to 28 days) and 23.5 (range: 15 to 36 days) days, respectively [41]. These results are probably due to the different amounts of larynx removed. In HG only the portion corresponding to the glottic plane is removed so that the laryngeal anatomy and its dynamics during the act of swallowing remain like the normal airway allowing a better management of salivary secretions, a good elevation of the larynx during swallowing, less postoperative edema, and better respiratory space.

Four studies out of 14 reported functional vocal results, making these data unreliable for drawing firm conclusions [5,6,8,9]. Regarding voice outcomes, we considered MPT and VHI. All of the studies showed similar values for MPT and VHI. Topaloglu et al. reported slightly better values than the other studies, which is likely explained by a longer follow-up time of their cohort of patients [8]. These values are subject to wide variability in the literature, probably due to the different time of parameter recording compared with surgery. We would need more standardized recording to compare results more truthfully.

Significant conclusions regarding the advantages and limitations of the HG cannot be drawn based on the literature data. However, to date, HG might be considered a viable surgical option for selected T1–T2 glottic cancers, whenever the laryngeal exposure is poor and/or the anterior commissure is involved, because of the outcomes of radiation therapy worse both in terms of DSS and OS. Moreover, it might be considered as a possible option in elderly patients as an alternative to OPHL type 2, to avoid extensive removal of laryngeal subsites with subsequent longer and tougher rehabilitation. Therefore, more studies are still needed to better understand the role of HG in different clinical scenarios.

This systematic review presents several limitations due to the lack of studies in the current literature focusing on HG. Furthermore, the variables reported herein evaluating the oncologic and functional results are quite heterogeneous due to the retrospective nature of most of the studies selected and the difference outcomes measured, not allowing for data pooling and metanalysis.

## 5. Conclusions

According to the results of this systematic review, HG is an oncologically safe surgical option for T1a–T1b glottic tumors with oncological outcomes at least comparable to other treatment options. HG could be a good option whenever poor laryngeal exposure and/or patient’s medical contraindications or refusal of radiotherapy are encountered.

As already mentioned, no certain conclusions emerge from this systematic review. However, having more treatment options available for early stage laryngeal cancers allows therapy to be modulated not only on the characteristics of the tumor but also on those of the patient and his preferences.

## Figures and Tables

**Figure 1 jcm-12-02261-f001:**
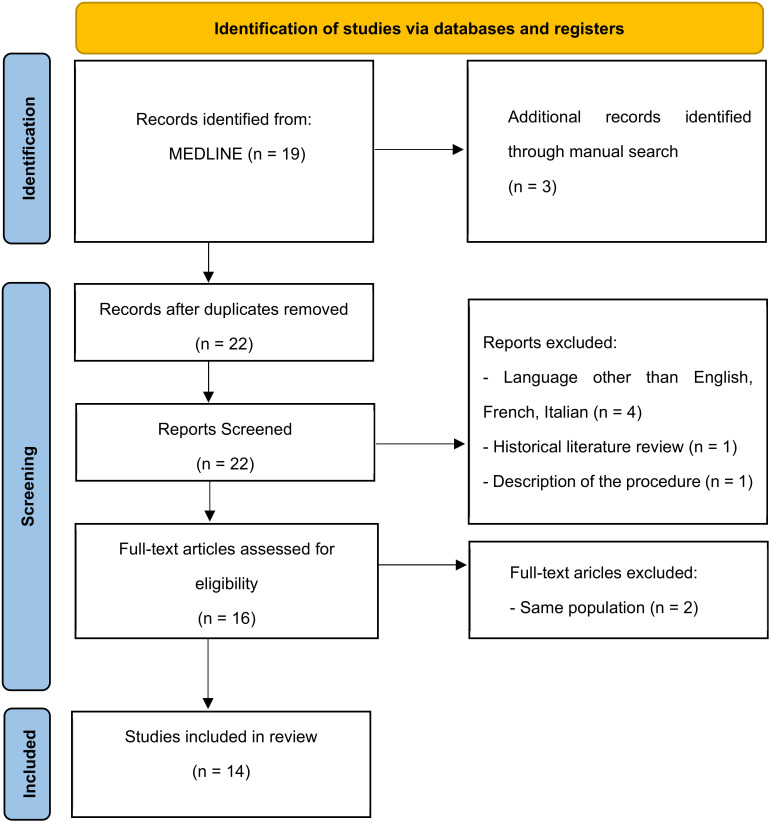
Flow chart of the review.

**Figure 2 jcm-12-02261-f002:**
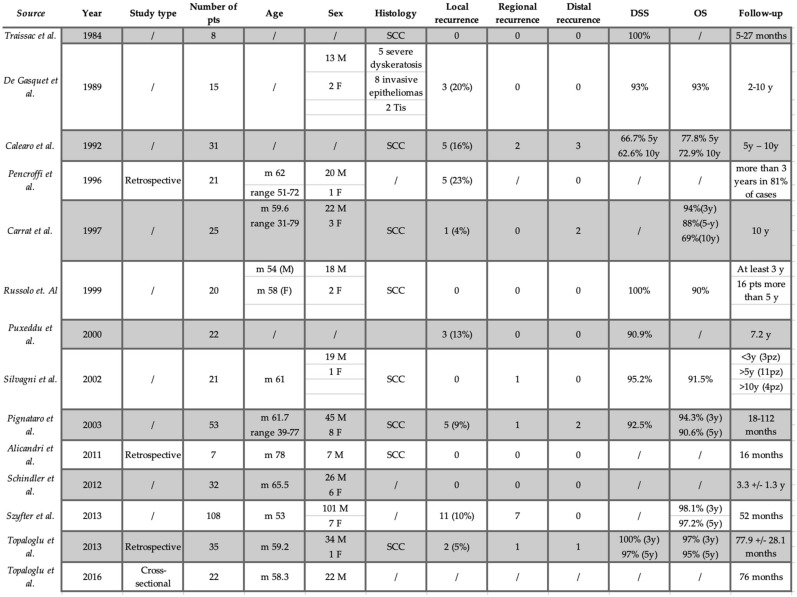
Oncological outcomes. M: male; F: female; y: year; pts: patients; m: median; OS: overall survival; DSS: disease-specific survival; SCC: squamous cell carcinoma.

**Table 1 jcm-12-02261-t001:** Tumor preoperative staging. N. = number of patients.

Author	N.	Preoperative Staging
L. Traissac, 1984	8	3 T1a–5 T1b
De Gasquet, 1989	15	15 T1b
Calearo, 1992	31	3 diffuse hyperkeratosis
		6 Tis
		14 T1a
		8 T1b
Pencroffi, 1996	21	21 T1b N0 M0
Carrat, 1997	25	13 T1a
		12 T1b
		N0
M. Russolo, 1999	20	15 T1b
		5 T1A
		N0
Puxeddu, 2000	22	Non
C. Silvagni, 2002	21	3 T1a
		18 T1b
		N0M0
Pignataro, 2003	53	53 T1b
		N0
Alicandri, 2011	7	Non
Antonio Schindler, 2012	32	Non
Witold Szyfter, 2013	108	108 T1b
		10 positivi al II livello
İlhan Topaloğlu, 2013	35	26 T1b N0
		2 T1b N1
		6 T2 N0
		1 T2 N1
İlhan Topaloğlu, 2016	22	16 T1b N0M0
		2 T1b N1M0
		3 T2 N0M0
		1 T2 N1M0

**Table 2 jcm-12-02261-t002:** Assessment of quality and risk of bias. MINORS: Methodological Index for Non-Randomized Studies.

Study	MINORS
**Traissac** et al.	6
**De Gasquet** et al.	4
**Calearo** et al.	9
**Pencroffi** et al.	7
**Carrat** et al.	9
**Russolo** et. al.	9
**Puxeddu** et al.	11
**Silvagni** et al.	11
**Pignataro** et al.	11
**Alicandri** et al.	11
**Schindler** et al.	11
**Szyfter** et al.	11
**Topaloglu** et al.	11
**Topaloglu** et al.	11

## Data Availability

No new data were created.

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
