# Peer review of "Oncological and Functional Outcomes for Horizontal Glottectomy: A Systematic Review"

_jcm, 2023, doi:10.3390/jcm12062261_

Round 1

Reviewer 1 Report

Thank you for submitting your article. It is an interesting topic, and I have the following comments/suggestions: 

Abstract:

1. Line 23 - "in 35, 12 and 8" cases? patients?

2. Line 26-29 - consider replacing the word "option" with a synonym. You use it three times in two sentences. 

3.  Line 29 - what do you mean by patient's impossibility? I do not know what it means. 

Introduction

4.  Please check spelling and punctuation across the manuscript. The missing comma is common in the manuscript. Example Line 48, after more specifically. 

5. Line 56-58. Explain shortly why this method lost its importance. What are the advantages of TLM and OPHL.

Materials and Methods:

6. In the flow chart you state that you added 3 articles manually. You do not mention it in Materials and Methods. Please correct it.

Results:

7. Line 108 – “consisting in SCC” – please rewrite.

8. Line 109 - ± missing between numbers

9. Line 115 – 116. Clarify, unclear.

10. Table needs a legend with an explanation of abbreviations. n. in table represents number of patients?

11. Line 129 – comma missing

12. Line 136 – reported instead of expressed?

13. Line 137-138 – change to “the mean VHI average was…”

14. Line 142-143 – I do not understand why it is in the text. Text in []. Are these your notes?

Discussion:

15. Is AC+ and ACi+ same thing? You use sometimes AC+ and sometimes ACi+. Confusing.

16. the correct spelling of the abbreviation “et al.” is with a dot. Correct across the manuscript.

17. Line 171-172 – rewrite for better English

18. Line 178 – local recurrence? Spelling

19. Line 183 – comma is missing.

20. Line 185 – level of AC?

21. Line 188 – worse local control?

22. Line 207-210 – you repeat the results. Remove this section.

23. Line 236-237 – clarify, rewrite. ”being the outcomes…”

Conclusions:

24. replace patient's impossibility?

Summary: Well done literature review, interesting topic. The weakest part of the manuscript is that there are a lot of spelling mistakes and sometimes it is difficult to guess what the authors want to say. However, it can be easily fixed.  A remark for the future. 8 authors for a review based on 14 manuscripts is a bit too many. One can question if they all have contributed enough to be co-authors. Consider it when designing future projects.

Reviewer 2 Report

As already mentioned in your conclusion that , no certain conclusions emerge from this systematic review other than that one available option of addressing the early disease by Horizontal glottectomy. Anterior commissure involvement requires more aggressive treatment and bigger fields including the drainage area while treating such patients with Radiotherapy. 

Author Response

REVIEWER #2

As already mentioned in your conclusion that, no certain conclusions emerge from this systematic review other than that one available option of addressing the early disease by Horizontal glottectomy. Anterior commissure involvement requires more aggressive treatment and bigger fields including the drainage area while treating such patients with Radiotherapy. 

Yes, agreed. However, as it is reported in the discussion subheading, there are still several patients and indications where the HG can be a viable option to achieve good oncologic and functional outcomes, sparing the radiation therapy treatment for possible second primary tumor or recurrent tumor.

Reviewer 3 Report

Dear authors, this is a well presented study on a systematic review on the outcomes of of horizontal glottectomy. Indeed, there are not a lot of studies in the literature and the conclusions of outcomes must be moderately examined. The presentation of the review is of good quality. I do not have any more comments.

Author Response

Dear authors, this is a well presented study on a systematic review on the outcomes of of horizontal glottectomy. Indeed, there are not a lot of studies in the literature and the conclusions of outcomes must be moderately examined. The presentation of the review is of good quality. I do not have any more comments.

Thank you for your valuable comments. We have obviously pointed out in the limitations’ subheading the drawbacks of this research projects. However, this is the first systematic review on the topic which is still of interest and can be useful for the reader to get the pooled results of a comprehensive literature review.

Reviewer 4 Report

In this study, the authors perform a systematic review of literature about oncological and functional outcomes for horizontal glottectomy. This topic is interesting however, some concerns should be addressed:

1-The systematic review was performed according PRISMA guidelines. It was registered in PROSPERO? In affirmative case, provides register number.

2-The search strategy should be described in Methods section. Please include the terms of search and the different terms combinations.

3- Flow Diagram should be performed using the PRISMA template.

4-The quality of studies was not evaluated. Authors should be evaluating the quality of the included studies and discuss these results.

5- A Table summarize the characteristics of the included studies should be performed

6- What´s mean AC? Please, include the significance in the manuscript.

7-There are some mistakes along the manuscript.

Line 82 (laryngeal squamous cell carcinoma): LSCC

Line 87 (laryngeal squamous cell carcinoma): LSCC

Line 178 (locar): local

Line 239 (mores): more

Author Response

REVIEWER #4

In this study, the authors perform a systematic review of literature about oncological and functional outcomes for horizontal glottectomy. This topic is interesting however, some concerns should be addressed:

1-The systematic review was performed according PRISMA guidelines. It was registered in PROSPERO? In affirmative case, provides register number.

No it was not registered in PROSPERO, but the authors preliminarly verified for the presence of similar systematic reviews on the topic and none of those was found.

2-The search strategy should be described in Methods section. Please include the terms of search and the different terms combinations.

The search strategy was reported in the methods sub heading and, being the name of the surgical intervention “horizontal glottectomy”, the authors started the PubMed and Scopus database search by typing this string. Moreover, we performed thorough manual search based on articles’ bibliographies to possibly achieve more records to the qualitative and quantitative analysis.

3- Flow Diagram should be performed using the PRISMA template.

The flow diagram was modified according to the PRISMA template, as suggested.

4-The quality of studies was not evaluated. Authors should be evaluating the quality of the included studies and discuss these results.

The quality of studies was evaluated according to MINORS criteria as it was added to the methods and results subheadings.

5- A Table summarize the characteristics of the included studies should be performed

As the reviewer requested, a table (table 2) was added to the results subheading reporting the main characteristics of the included studies and a resume of the main oncologic outcomes.

6- What´s mean AC? Please, include the significance in the manuscript.

The meaning of AC was specified in the line 48.

7-There are some mistakes along the manuscript.

Line 82 (laryngeal squamous cell carcinoma): LSCC

Line 87 (laryngeal squamous cell carcinoma): LSCC

Line 178 (locar): local

Line 239 (mores): more

Ok. These mistakes have been addressed.

Round 2

Reviewer 4 Report

The minor revisions were carried out according to the suggestions.